# Determinants of Clinical Remission in Dupilumab-Treated Severe Eosinophilic Asthma: A Real-World Retrospective Study

**DOI:** 10.3390/biomedicines13102404

**Published:** 2025-09-30

**Authors:** Matteo Bonato, Elisabetta Favero, Francesca Savoia, Matteo Drigo, Simone Rizzato, Enrico Orzes, Gianenrico Senna, Micaela Romagnoli

**Affiliations:** 1Pulmonology Unit, Ca’ Foncello Hospital, Azienda Unità Locale Socio-Sanitaria 2 Marca Trevigiana (AULSS2), 31100 Treviso, Italymicaela.romagnoli@aulss2.veneto.it (M.R.); 2Internal Medicine 1, Ca’ Foncello Hospital, Azienda Unità Locale Socio-Sanitaria 2 Marca Trevigiana (AULSS2), 31100 Treviso, Italy; 3Department of Internal Medicine, University of Verona, 37134 Verona, Italy

**Keywords:** CRSwNP, GERD, obesity, hypertension, bronchiectasis, biologics, nasal polyps, FEF_25–75_

## Abstract

**Background**: Dupilumab, a monoclonal antibody targeting the IL-4/IL-13 receptor, has shown significant efficacy in improving asthma control and reducing exacerbations in patients with severe eosinophilic asthma. However, there is a lack of knowledge about real-world data on clinical remission rates and their predictors. **Objective**: This study aimed to evaluate clinical outcomes, remission rates, and predictive factors of remission in a real-life cohort of patients with severe eosinophilic asthma treated with dupilumab. **Methods**: We conducted a retrospective, bicentric, observational study including 52 patients with severe eosinophilic asthma treated with dupilumab. Clinical, functional, and biomarkers were assessed at baseline, 6 months, and 12 months. Statistical analyses included logistic regression to identify independent predictors of clinical remission. **Results**: After 12 months of treatment, 48.2% of patients achieved clinical remission. Dupilumab significantly improved asthma control and lung function (including FVC and FEF_25–75_), reduced exacerbation rates, and maintenance therapy. High blood eosinophil levels (>490 cells/µL), high FeNO levels (>25 ppb), a history of CRSwNP, and better baseline FEV1 were associated with asthma remission. Conversely, obesity (BMI > 30) and related comorbidities (such as GERD, OSAS, and hypertension) and bronchiectasis were associated with a lower likelihood of remission. Multivariate analysis confirmed higher baseline FEV1 (OR 2.94; IC 1.13–7.6), positive history of CRSwNP (OR 8.03; IC 1.41–45.5), and higher baseline blood eosinophils (OR 1.003 IC 1.001–1.006) as independent predictors of clinical remission. **Conclusions**: These results are in line with the known effectiveness of dupilumab in severe eosinophilic asthma and identified a history of CRSwNP, higher baseline FEV1, and elevated blood eosinophils as key predictors of clinical remission. These findings may contribute to a more personalized approach to treatment selection, emphasizing the importance of comorbidity assessment together with type 2 inflammation biomarkers. Further prospective studies are needed to validate these results.

## 1. Introduction

Asthma is a common chronic airway disease that affects around 300 million people worldwide, with about 3.7% experiencing severe asthma [1,2]. Severe asthma significantly impairs patients’ quality of life and accounts for nearly 60% of the overall economic burden of asthma [1,2]. T2 high is the prevalent inflammatory phenotype among patients with severe asthma, including 70–89% of patients [3]. Over the past decade, biologic therapies have proven to be both safe and effective in improving outcomes and the quality of life for patients with severe T2-high asthma. Dupilumab, a monoclonal antibody targeting IL4/IL13 receptor, has demonstrated its efficacy since 2018 in improving asthma control, lung function, and reducing asthma exacerbation and OCS-dependence in asthmatics with type 2 inflammation [4].

Despite these results, further real-life evidence is needed, particularly on clinical remission as the key measure of overall response to dupilumab. Additionally, there is a need to identify additional predictors of clinical response (or non-response) to the drug, beyond the already known type 2 inflammation markers. Indeed, clinical and biomarkers predicting outcome in dupilumab-treated patients emerged from the literature (such as corticosteroid dependence, high FeNO levels, and elevated blood eosinophil counts) are similar to all other biologics for severe asthma: current evidence is actually insufficient to guide drug choice in patients eligible for more than one biologic drug [1,2].

In this study, we aim to evaluate the real-life effectiveness of dupilumab in patients with severe eosinophilic asthma. The primary objective is to identify clinical, pulmonary function, and biomarkers at baseline that are associated with asthma remission after 12 months of treatment. In addition to this, the study seeks to assess the clinical outcomes at 6 and 12 months of therapy in a real-life context.

## 2. Materials and Methods

### 2.1. Study Design

An observational, real-life, retrospective, bicentric study was carried out to analyze the clinical records of patients who started dupilumab for severe asthma (n = 52) from 1 January 2022 to 31 December 2023 in severe asthma outpatient clinics of two centers (Treviso and Vittorio Veneto City hospitals, Italy). The study has been approved by the local ethical committee (ref. 1307/CE Marca, RINOVA).

### 2.2. Population

All patients had an asthma diagnosis according to GINA guidelines [1] and severe asthma according to the ERS/ATS criteria [5]. All patients received dupilumab for severe asthma as the first indication, and they were >18 years old at treatment start. The patients in the study could be either treatment-naive or switched to dupilumab from a previous therapy. All patients treated complied with national indications for dupilumab prescription (see Appendix A). The decision to treat patients with dupilumab instead of other biologics was made by a physician expert in severe asthma management, according to his judgment.

### 2.3. Patient Evaluation and Variables Assessed

A schematic flow chart of the study design and visit assessments is provided in Figure 1. All severe asthma patients eligible for biologic therapy underwent a baseline visit (V0) within 30 days prior to the first dose of dupilumab, and then, every 6 months thereafter as follow-up visits. Clinical data, pulmonary function, and biomarkers at treatment start (V0), at 6 months (V1, mean follow-up 5.2 ± 2.2), and at 12 months (V2, mean follow-up 12.0 ± 5.3) of treatment were extrapolated from the electronic patients’ registry. Data considered included: asthma control test (ACT) result, acute exacerbation (AE) rate, severe acute exacerbation rate, maintenance antiasthmatic treatment, blood eosinophil count, and respiratory function tests (performed according to the ERS/ATS guidelines [6]. Allergological test (total and specific serum IgE dosage or prick test) and nitric oxide dosage on exhalation have been performed only at V0. Comorbidities and clinical past history have also been recorded. More details on variable descriptions are reported in the Appendix A.

Due to the observational and real-life design of the study, patients were evaluated according to the internal protocol of our severe asthma centers. No additional tests or evaluations were performed for research purposes. In our study, all data (with the exception of FeNO, which was performed in only 22 out of 52 patients) were available for the entire cohort of patients (n = 52) at each follow-up visit. Clinical remission was defined at V2, according to Menzies-Gow criteria [7], and operatively applicated as follows: (a) sustained absence of significant asthma symptoms based on validated instrument, defined as ACT score ≥ 20 at T2; (b) optimization and stabilization of lung function, defined as a normalization of lung function (FEV1 > 80% of predicted value) according to Hansen S. et al. [8]; (c) patient and healthcare provider agreement regarding remission; and (d) no use of systemic corticosteroids therapy for exacerbations treatment or long-term disease control.

### 2.4. Statistical Analysis

Statistical analysis description was reported in the Appendix A.

## 3. Results

### 3.1. Study Population

Fifty-two patients were included in the study. Of them, 31 (59.7%) have not been previously treated with biologics for asthma (naive), while 21 (40.3%) were switched from another biologic: 10 (19.2%) from omalizumab, 9 (17.3%) from mepolizumab, and 2 (3.8%) from benralizumab. Comparisons of baseline clinical characteristics between biologic-naïve and switched patients are presented in the Appendix A, and no significant differences were observed between the two groups. None of the fifty-two patients discontinued dupilumab during the study. At baseline, the mean age was 53.9 ± 17.1 years, and male prevalence was 51.9%. In the cohort, there were no smokers, but 22 (40.7%) patients were former smokers. Atopic patients were 39 (75%) and, among them, 29 (55%) were sensitized to perennial allergens, while 10 (19%) were exclusively sensitized to seasonal allergens. Asthma onset was early (<18 years) in 22 patients (42.3%) and late (>40 years) in 16 (30.7%). The most frequent comorbidities were the following: allergic rhinitis (n = 36, 69.2%), followed by CRSwNP (n = 30, 57.6%), GERD (n = 25, 48%), atopic dermatitis (n = 13, 25%), obesity (n = 11, 21.1%), hypertension (n = 9, 17.3%), OSAS (n = 7, 13.4%), bronchiectasies (n = 7, 13.4%), type 2 diabetes (n = 4, 7.6%), ACO (n = 3, 5.7%), and major depression and active cancer (both n = 2, 3.8%). Other comorbidities considered (ABPA and EGPA) have not been observed in the cohort. The levels of the three assessed biomarkers at baseline were significantly above the level of normality, in particular, blood eosinophils mean was 457 ± 332 cell/µL (v.n. 0–250 cell/µL), total serum IgE 926 ± 1578 kU/L (v.n. 0–114 kU/L) and FeNO 51.5 ± 39.2 ppb (v.n. < 25 ppb).

### 3.2. Clinical and Functional Outcome at 6- and 12-Month Follow-Up

Clinical characteristics at baseline (V0), 6 months (V1), and 12 months (V2) follow-up are reported in Table 1. A significant improvement of almost all clinical and functional parameters was observed in the cohort. As reported in Figure 2A, a significant improvement of ACT score was observed at V1 (mean: +4.0 ± 5.5 pts; *p* < 0.0001) and V2 follow-up (mean: +5.4 ± 5.4 pts vs. V0; *p* < 0.0001 and +1.3 ± 1.4 vs. V1; *p* = 0.005). A significant decrease in acute exacerbation rate has been observed at 6 and 12 months (mean: −31.6 ± 100%; *p* = 0.01 at 6 months and −40.8 ± 76.6%; *p* = 0.004 at 12 months, Figure 2B), and of severe acute exacerbations both at 6 and 12 months (mean:−92.8 ± 27.7%; *p* = 0.003 at V1 and −100% at V2; *p* = 0.001, Figure 2C). Of note, no more severe acute exacerbations have been documented in the cohort after the first 6 months of treatment.

As shown in Figure 2D,E, a significant functional improvement of the airflow obstruction was already observed at V1 and was maintained at 12 months (FEV1 mean: +261 ± 444 mL; *p* < 0.0001 at V1 and +256 ± 552 mL; *p* = 0.02 at V2; FEF_25–75_ mean: +392 ± 573; *p* = 0.0006 at V1 and +213 ± 1084 mL; *p* = 0.001 at V2). No significant further improvement was observed in pulmonary function between V1 and V2. Notably, a significant improvement was also observed in FVC and the FEV1/FVC ratio (see Table 1).

Figure 3 shows maintenance anti-asthmatic treatment prevalence at the three timepoints. A significant reduction in maintenance treatment was observed from V0 and V1 (−13.3%; *p* < 0.0001 for high-dose ICS; −37.5%; *p* = 0.0002 for OCS; −18.5%; *p* < 0.0001 for LAMA) and further from V1 and V2 (−18.9%; *p* = 0.004 vs. V1 and −31.8%; *p* < 0.0001 vs. V0 for high-dose ICS; −66.6%; *p* = 0.002 vs. V1 and −79.1%; *p* = 0.011 vs. V0 for OCS; −27.2%; *p* < 0.0001 vs. V1 and −40.7%; *p* < 0.0001 vs. V0 for LAMA, Table 1). Of note, among patients who were still in maintenance OCS treatment, the daily OCS dose also significantly decreased from V0 to V1 (−62.1 ± 46%; *p* = 0.001) and further from V1 to V2 (−89.7 ± 24.7%; *p* < 0.0001 vs. V0 and −73.2 ± 42.1%; *p* = 0.003 vs. V1, Figure 2F). As illustrated in Table 1, blood eosinophils almost doubled from V0 to V1; nevertheless, the difference was not statistically significant, probably due to the lack of standardization in the timing of the samples.

Given the importance of atopic asthma as a distinct sub-endotype of T2 high asthma, history of smoking, and of different previous biologic anti-asthmatic treatment as a potential confounding factor, we assessed a stratification of the results by these three variables. In this cohort, atopy, smoking history, and previous biological treatment for asthma did not significantly influence the improvement in clinical and functional variables at 6 and 12 months. Moreover, in this cohort, baseline total serum IgE did not significantly predict greater clinical or functional benefit from dupilumab between V0 and V2.

### 3.3. Asthma Remission Rate and Determinants of Clinical Remission at 12 Months

Figure 4A shows the prevalence distribution of the number of Menzies-Gow criteria achieved by each patient at 12 months. At 12 months in this cohort, 25 (48.2%) out of 52 patients achieved clinical remission (4 out of 4 criteria). Among patients who did not achieve remission, 12 (23.1% of the whole cohort) patients reached 3 out of 4 criteria, 10 (19.2%) patients achieved 2 out of 4 criteria, 3 (5.7%) patients achieved 1 out of 4 criteria, and 2 patients (3.8%) achieved zero criteria. Patients who achieved less than two criteria have been defined as non-responder; these patients were considered as part of the non-remission group in the following analyses.

Clinical, respiratory functional, and levels of biomarkers at baseline associated with asthma remission after 12 months of dupilumab treatment are summarized in Table 2. Neither age nor sex, nor smoke status, nor asthma onset age were significantly associated with asthma remissions at 12 months follow-up. The prevalence of atopy, allergic rhinitis, and atopic dermatitis did not differ between the two groups, while a significantly higher prevalence of patients exclusively sensitized to seasonal allergen (SAS) among the remission group (*p* = 0.027; Figure 4B) was noted.

Among comorbidities (Figure 4B), CRSwNP was associated with a higher likelihood of asthma remission (*p* = 0.001). Conversely, patients who achieved asthma remission at 12 months had a lower prevalence of GERD (*p* = 0.024), obesity (*p* = 0.0003), OSAS (*p* = 0.043), hypertension (*p* = 0.014), and bronchiectasis (*p* = 0.043). Notably, no obese patients were observed in the remission group. Patients who achieved asthma remission at 12 months showed at baseline better pulmonary function parameters (*p* = 0.042 for FVC; *p* = 0.004 for FEV1, Figure 4C; *p* = 0.0007 for FEV1/FVC and *p* = 0.002 for FEF_25–75_, Figure 4D). ACT symptoms scale, the rate of previous acute exacerbations, and maintenance antiasthmatic treatment at baseline were not significantly associated with asthma remission at follow-up.

Among biomarkers, baseline blood eosinophils (*p* = 0.008; Figure 4E) and FeNO > 25 ppb (*p* = 0.032; Figure 4B) but not total serum IgE were significantly associated with asthma remission at 12 months. Limiting this analysis to patients who had never been previously treated with other biologics for asthma, this significant association between asthma remission and higher blood eosinophil levels (574 ± 336 vs. 272 ± 191 cell/µL; *p* = 0.013) or FeNO > 25 ppb (75% vs. 14,2%; *p* = 0.0406) at baseline was confirmed.

Among baseline continuous variables significantly associated with asthma remission at V2, an ROC analysis has been performed to assess the diagnostic accuracy of blood eosinophils and pulmonary function in naive patients to predict asthma remission after 12 months of dupilumab treatment. Diagnostic accuracy was overall acceptable only for blood eosinophils with an AUC of 0.698, a sensitivity of 60%, and a specificity of 87% at the best threshold of 490 cells/µL (Figure 4F).

Based on these results, further analyses were conducted to explore potential factors that could have influenced the observed outcomes. As supposed, obesity was significantly associated with GERD, OSAS, and hypertension (*p* = 0.012; *p* < 0.0001; *p* = 0.0002, respectively) but not with CRSwNP, bronchiectasies, and perennial allergen sensitization. Additionally, obese patients demonstrated more compromised baseline pulmonary function compared to their non-obese counterparts (FEV1 2.54 ± 0.958 L vs. 1.827 ± 0.230 L; *p* = 0.031). Similarly, patients sensitized to perennial allergens had more impaired baseline pulmonary function than those exclusively sensitized to seasonal allergens (FEV1 2.04 ± 0.853 L vs. 2.902 ± 0.943 L; *p* = 0.003). Conversely, CRSwNP and bronchiectasis patients did not show more impaired pulmonary function at baseline. Regarding biomarkers, no significant differences in baseline levels of blood eosinophils or FeNO were observed across any of the variables considered.

These findings were further supported by the results of a multivariate logistic regression analysis, which confirmed that a positive history of CRSwNP (OR 8.03, CI 1.41–45; *p* = 0.018), a higher blood eosinophil count (OR 1.003, CI 1–1.006; *p* = 0.015), and higher baseline FEV1 (OR 2.73, CI 1.19–6.70; *p* = 0.026) were independently associated with asthma remission after 12 months (Table 3).

## 4. Discussion

This retrospective real-life bicenter study, focused on a cohort of severe asthmatic patients treated by dupilumab, provides several novel insights into asthma remission. First, it validates previously reported predictors of remission within a single, homogeneous cohort of patients treated exclusively with dupilumab. Second, it reports the association between obesity and poor therapeutic response to encompass other obesity-related comorbidities, including GERD, OSAS, and hypertension. Third, the study suggests the presence of bronchiectasis as a novel negative predictor of therapeutic response to biologic therapy in asthma. Fourth, this study may support the efficacy of dupilumab on improving FVC and FEF_25–75_, suggesting a potential beneficial effect also on distal airways, although an oscillometer was not used in this study. Finally, in this study, we observed not only a reduction in OCS but also a significant decrease in high-dose ICS and LAMA inhaled therapies in patients treated with dupilumab, which represents a novelty.

This study suggests that dupilumab efficacy is provided already after the first 6 months from treatment start, and about improving both asthma symptoms and respiratory function, reducing both exacerbation rate and maintenance treatment. On this point, we observed a clinical remission rate of 48.2%, which we considered in line with clinical trials [9] and most real-life studies [10,11,12,13,14]. Precisely, the benefit of dupilumab on respiratory symptoms, pulmonary function improvement, and acute exacerbation rate reduction was in line with other real-life cohorts [10,11,12,13,14,15] and clinical trials [16,17,18,19]. On the contrary, we observed a decrease in maintenance asthma treatment that appeared to be greater than that reported in the existing literature. More in detail, seventy-nine percent of patients have controlled asthma, weaned from OCS maintenance treatment after 12 months, a result superior to both trials [17] and other real-life studies [10,11,12]. Furthermore, we also found a significant decrease in the number of patients treated with high doses of ICS and with LAMA, indicating that a good response to biologic therapy may allow a step-down of inhaled therapy as well, as proved for treatment with benralizumab [20,21,22]. Indeed, the need for maintenance inhaled therapy with low-to-medium doses of inhaled corticosteroids (ICS) is considered one of the criteria for defining remission, according to a recent American consensus [23]

Moreover, this study evaluated other respiratory functional parameters, demonstrating a significant improvement in FVC and FEF_25–75_, suggesting a possible beneficial action of dupilumab also on the peripheral airways, resulting in a reduction in alveolar air-trapping. Indeed, it is established that interleukin-13 exerts a direct action on the bronchial wall by inducing hyperplasia of both the bronchial smooth muscle and the goblet cells, as illustrated in Figure 5 [4]. This result is in line with the premises of the recent VESTIGE study and its end-points [18].

In this cohort, patients who achieved clinical remission after 12 months of treatment had a positive history of chronic rhinosinusitis with nasal polyposis, allergic sensitization exclusively to seasonal allergens, a better baseline respiratory function, and high levels of blood eosinophils and nitric oxide on exhaled air. Conversely, a positive history of allergic sensitization to perennial allergens, obesity, GERD, OSAS, and hypertension was associated with a higher likelihood of non-remission.

As known from the literature [24], in this cohort, dupilumab has been shown to be equally effective in atopic and non-atopic patients, and baseline total serum IgE levels did not influence remission likelihood at 1-year follow-up. However, the percentage of patients who achieved remission was significantly higher in patients sensitized exclusively to seasonal allergens. This result might be influenced by the wide functional impairment at baseline of subjects sensitized to perennial allergens; indeed, this result was not confirmed by multivariate analysis. Therefore, our results are consistent with previous evidence of dupilumab effectiveness in severe asthma, regardless of atopy and serum total IgE levels. This may suggest that the IL-4/IL-13 inflammatory pathway is relevant in both non-allergic and allergic asthma, since IL-13 can also be released independently of allergen-driven responses by innate lymphoid cells type 2, as illustrated in Figure 5 [4].

Similar to findings from RCT [16,17,18,19] and real-life studies [10,11,12,13,14,15], blood eosinophils and FeNO were significantly associated with a good response to therapy and clinical remission. Notably, in this real-life cohort, all patients exhibited elevated baseline levels of blood eosinophils and/or FeNO, as required by national indications for dupilumab prescription. This might reinforce the potential existence of a quantitative relationship between treatment response and the intensity of type 2 (T2) inflammation. Although nitric oxide is a well-established direct marker of IL-4/IL-13 activity, as IL-13 stimulates the expression of inducible nitric oxide synthase in the bronchial epithelium, dupilumab does not directly target eosinophils by interfering with IL-5 production. Instead, IL-4 and IL-13 indirectly modulate eosinophilic inflammation by upregulating adhesion molecules such as VCAM-1 and ICAM-1, which facilitate eosinophil migration from the bloodstream into the airways, as illustrated in Figure 5 [4].

Furthermore, this study identified a threshold value of 490 cells/μL for blood eosinophils that predicted clinical remission with moderate accuracy, which might have practical implications in clinical decision-making. This threshold is higher than the commonly used cut-off of 300 cells/μL derived from clinical trials, yet aligns more closely with the 450 cells/μL value we previously identified as predictive of tissue eosinophilia in bronchial biopsies [25]. These findings further support peripheral blood eosinophils as a biomarker with high positive predictive value, but limited negative predictive value, probably because of the non-linear correlation between tissue and peripheral eosinophilia, as they represent distinct biological compartments.

Another element closely associated with the T2-high endotype is the presence of chronic rhinosinusitis with nasal polyposis, which, also in this study, appears to be an independent predictor of clinical remission. A significant association between clinical remission and CRSwNP has been demonstrated in the literature on cohorts including all biologics [26], but only in one study that included exclusively dupilumab-treated patients [27]. This result appears, in our view, to be explained by the pathophysiological continuity between the upper and lower airways, suggesting that the same type 2 immune mechanisms play a role both in the upper and lower respiratory tracts [28].

In this study, patients who achieved clinical remission at 12 months had better baseline lung function. Although this finding may have been influenced by the functional criteria used to define remission (FEV1 > 80%), we believe it deserves importance. Firstly, it has been replicated by only one other study in the literature [8]. Secondly, the result is not limited to the degree of obstruction (FEV1), but extends—to our knowledge, for the first time—to other lung volumes (FVC and FEF_25–75_). This finding could suggest a potential role of lung function in phenotyping severe asthmatic patients, as a quantitative surrogate marker of the severity of chronic bronchial inflammation and remodeling.

In this cohort, patients who achieved clinical remission after 12 months of treatment had a lower prevalence of bronchiectasis, obesity, and obesity-related comorbidities such as GERD, OSAS, and hypertension. A negative association between obesity and response to biologic therapy has already been confirmed in real-life cohorts of severe asthma patients treated with biologic drugs [8,26,29,30] and specifically with dupilumab [10]. However, to the best of our knowledge, no study has yet extended this negative association to other obesity-related comorbidities. Similarly, this study appears to be the first to identify a positive history of bronchiectasis as a negative predictor of clinical remission.

These comorbidities were not found to be independent predictors in the multivariate analysis, probably because they might themselves contribute to worse respiratory symptoms and impaired lung function, preventing the achievement of clinical remission either on clinical or functional criteria. Alternatively, it is conceivable that these patients could have an endotype of asthma that is not primarily T2-driven. Indeed, in obese asthmatic subjects, a neutrophilic pattern of inflammation—both bronchial and systemic—has been previously demonstrated [31,32], suggesting the involvement of non-T2-type pathways such as adipocites-derived IL-6 in the pathogenesis of airway inflammation.

This study has several limitations: first, the sample size, which is actually relatively limited, but it can be considered acceptable when compared to the cohorts reported in previous studies on dupilumab. Second, the retrospective design limits the predictive value of results by definition. Similarly, the observational real-life design limits the good quality of the data by definition, since it is less standardized and, in some cases, due to missing data (i.e., for nitric oxide). Third, the remission definition according to Menzies-Gow criteria, even widely supported by the literature, is itself limited by a lack of sensitivity. Indeed, the respiratory functional item (FEV1 > 80% pred.) excludes from the remission definition all those patients with baseline respiratory function characterized by an irreversible obstructive syndrome. These patients would unlikely be able to achieve “clinical remission”, despite the absence of respiratory symptoms and acute exacerbations due to treatment. Fourth, this cohort included either naive or previously biologic-treated patients, and this might have influenced the results, although we demonstrated no substantial difference between the two groups.

In conclusion, these results are in line with the known effectiveness of dupilumab in severe eosinophilic asthma, highlighting both known and novel potential predictors of therapeutic response. In particular, these results validated all known predictors, including a positive history of CRSwNP, higher baseline FEV1, and blood eosinophils within a single, homogeneous population, and suggest new potential functional and clinical determinants of clinical remission, such as small-airways pulmonary function, bronchiectasis, and obesity-related comorbidities. These findings may contribute to the personalized approach to biologic therapy in severe asthma, based on comorbidities, inflammatory profile, and lung function.

## Figures and Tables

**Figure 1 biomedicines-13-02404-f001:**
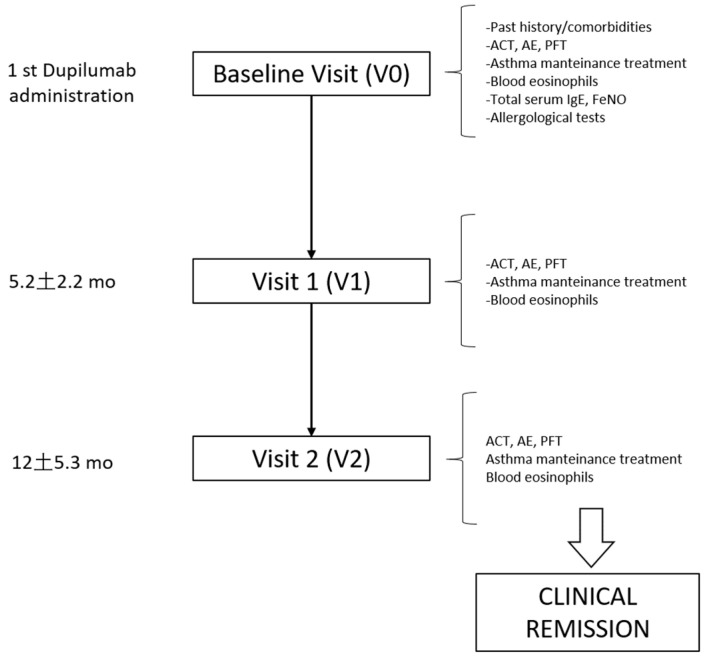
Design of the study. Flow chart reporting the study design and variable assessment at baseline and follow-up visits. ACT = asthma control test, AE = acute exacerbations, PFT = pulmonary function test.

**Figure 2 biomedicines-13-02404-f002:**
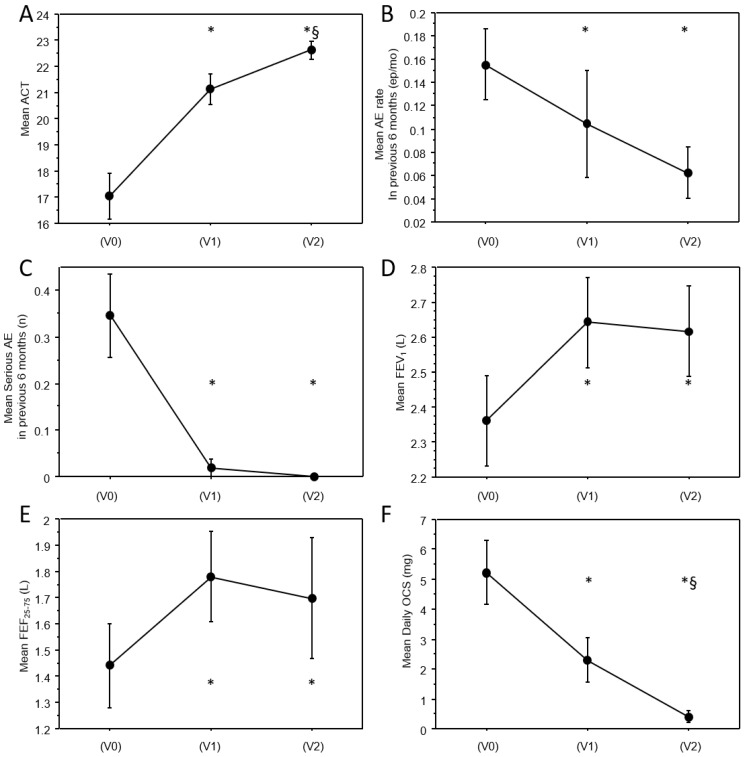
Trajectories of clinical and functional variables across study visits. Line charts representing the values of clinical and functional continuous variables at baseline and at follow-up visits, in particular: (**A**) mean ACT; (**B**) mean AE rate in previous six months; (**C**) mean serious AE in previous six months, (**D**) mean FEV_1_; (**E**) mean FEF_25–75_; (**F**) mean daily OCS. Vertical bars are standard errors. * significantly different from V0; §: significantly different from V1. *p*-values are reported in Table 1. AE = acute exacerbations.

**Figure 3 biomedicines-13-02404-f003:**
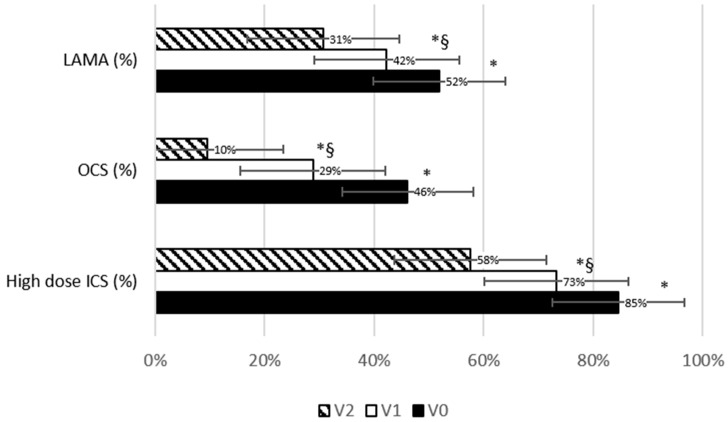
Trend in maintenance therapy prevalence across study visits. Bars chart representing the prevalence of maintenance anti-asthmatic therapies at baseline and at follow-up visits. The vertical bars represent the standard error. * significantly different from V0; §: significantly different from V1. *p*-values are reported in Table 1.

**Figure 4 biomedicines-13-02404-f004:**
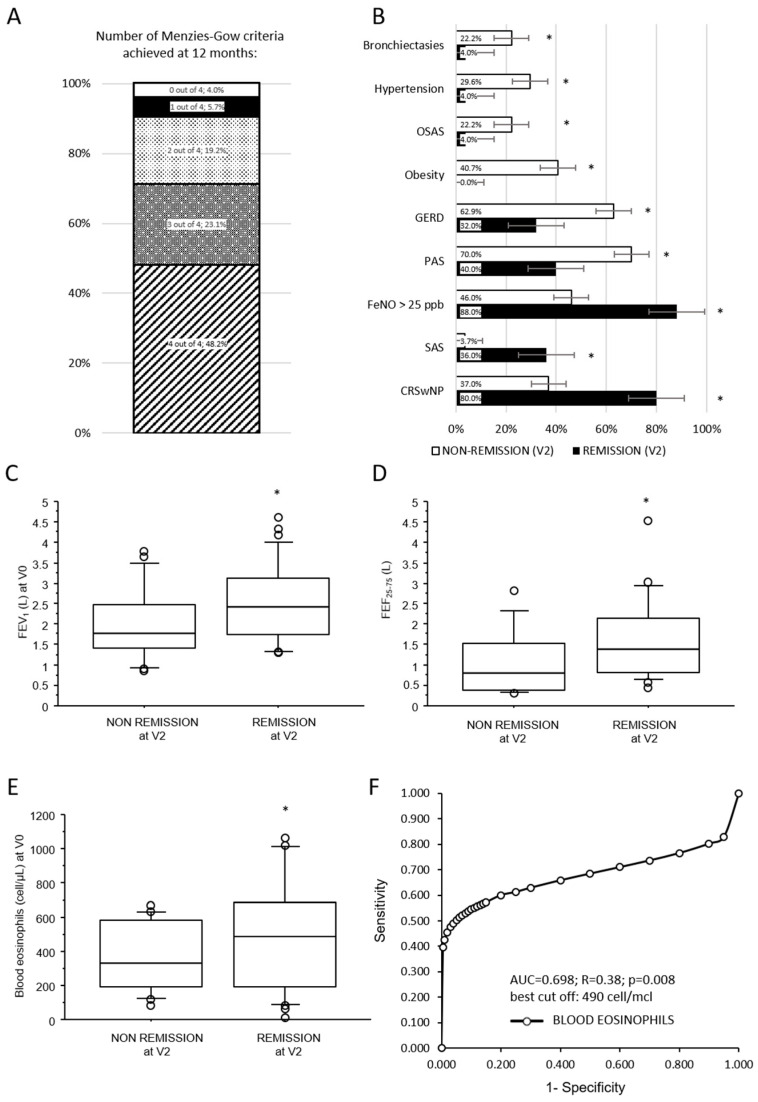
Rate and determinants of asthma clinical remission 12 months after dupilumab initiation. (**A**) Bar chart representing the distribution of the cohort by the number of Menzies-Gow criteria achieved at follow-up. (**B**) Bars chart representing prevalence of comorbidities in remission and non-remission patients after 12 months of dupilumab treatment; (**C**–**E**) box plots illustrating the comparison between baseline continuous variables and asthma remission after 12 months of dupilumab treatment. Solid line represents the median; bottom and top of the boxes are the 25th and 75th percentiles; brackets correspond to the 10th and the 90th percentiles. (**F**) Receiver operating characteristic curve of diagnostic accuracy of blood eosinophils at baseline to predict asthma remission after 12 months of treatment with dupilumab. A-axis reports 1-specificity and y-axis the sensitivity at progressive values of blood eosinophils. Best cut-off, sensitivity, and specificity are reported in the text. * *p* < 0.05; *p*-values are reported in Table 2. SAS = exclusively seasonal allergen sensitization; PAS = perennial allergen sensitization.

**Figure 5 biomedicines-13-02404-f005:**
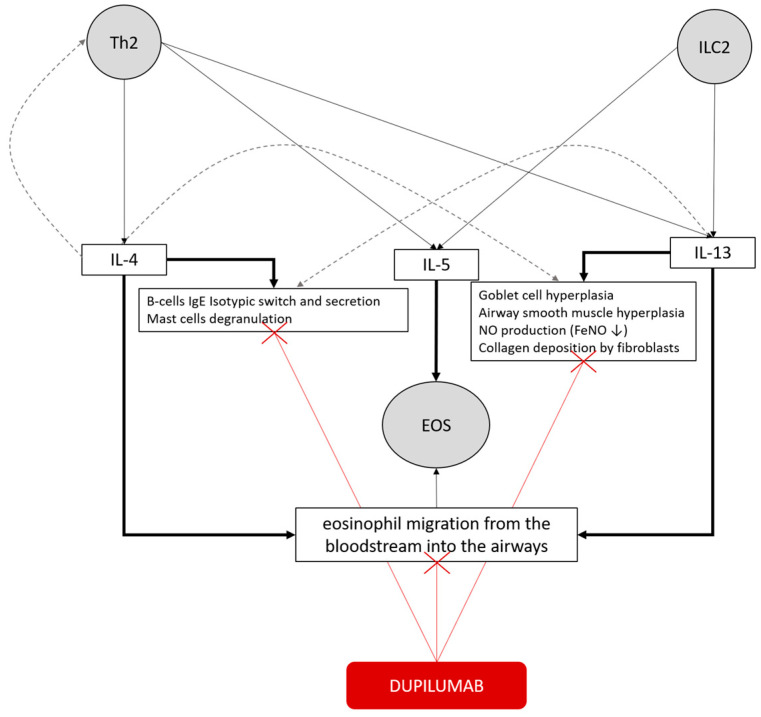
Concepts map of the type 2 inflammatory pathway in bronchial asthma and the mechanism of action of dupilumab. T helper 2 (Th2) cells and type 2 innate lymphoid cells (ILC2) release IL-4, IL-5, and IL-13. IL-4 primarily promotes B-cell IgE class switching and mast cell degranulation, whereas IL-5 drives eosinophil recruitment and survival. IL-13 induces goblet cell hyperplasia, airway smooth muscle hyperplasia, collagen deposition by fibroblasts, and FeNO production. These cytokines collectively sustain eosinophilic inflammation and promote eosinophil migration from the bloodstream into the airways. Redundancy is illustrated by gray dashed lines. Dupilumab, by blocking the IL-4 receptor α (IL-4Rα) subunit, inhibits IL-4 and IL-13 signaling, thereby reducing type 2 inflammation and its downstream effects (red lines).

**Table 1 biomedicines-13-02404-t001:** Comparison of clinical–functional characteristics and biomarkers across study visits: baseline (V0), 6 months (V1), and 12 months (V2).

	V0	V1	V2	*p*-Value (V0 vs. V1)	*p*-Value (V0 vs. V2)	*p*-Value (V1 vs. V2)
Visit distances (mo)	na	5.2 ± 2.2	12.0 ± 5.3	-	-	-
ACT (pts)	17 ± 6	21 ± 4	23 ± 3	<0.0001	<0.0001	0.005
AE rate (ep/months)	0.15 ± 0.29	0.11 ± 0.33	0.06 ± 0.15	0.010	0.004	n.s.
Severe AE (n)	0.34 ± 0.65	0.02 ± 0.14	0	0.003	0.001	n.s.
FVC (L)	3.610 ± 1.140	3.810 ± 1.099	3.750 ± 1.130	0.046	n.s.	n.s.
FEV1 (L)	2.360 ± 0.930	2.640 ± 0.913	2.620 ± 0.940	<0.0001	0.002	n.s.
FEF_25–75_ (L)	1.440 ± 1.070	1.781 ± 1.070	1.690 ± 1.204	0.0006	0.01	n.s.
FEV1/FVC	64.4 ± 12	68.8 ± 11.7	68.2 ± 11.1	<0.0001	0.003	n.s.
Blood eosinophils (cell/µL)	457 ± 332	808 ± 1247	738 ± 645	n.s.	n.s.	n.s.
PRIST (kU/L)	926 ± 1578	n.a.	n.a.	n.a.	n.a.	n.a.
FeNO (ppb)	51.5 ± 39.2	n.a.	n.a.	n.a.	n.a.	n.a.
High-dose ICS, n (%)	44 (84.6)	38 (73.3)	30 (57.6)	<0.0001	<0.0001	0.004
OCS treatment, n (%)	24 (46.1)	15 (28.8)	5 (9.6)	0.0002	0.011	0.002
Daily OCS dose (mg)	5.2 ± 7.8	2.3 ± 5.3	0.4 ± 3.5	0.001	<0.0001	0.003
LAMA, n (%)	27 (51.9)	22 (42.3)	16 (30.7)	<0.0001	<0.0001	<0.0001

Data are reported as mean ± standard deviation for continuous variables or absolute (relative) frequency for nominal variables. The comparison between baseline and follow-up at 6 and 12 months has been performed with Wilcoxon signed rank test for continuous variables or chi-square/Fisher’s exact test for nominal variables. n.s. = not significant; n.a. = not applicable. Abbreviations are reported in the Materials and Methods Section. Definitions of total and severe acute exacerbations (AE) are also reported in the Materials and Methods Section.

**Table 2 biomedicines-13-02404-t002:** Comparison of clinical–functional characteristics and biomarkers according to asthma clinical remission 12 months after dupilumab initiation.

	Remission at 12 Months	Non-Remission at 12 Months	*p*-Value
Subjects (n)	25	27	-
Age (years)	53.9 ± 20	53.5 ± 12.5	n.s.
Males, n (%)	15 (60)	12 (44.4)	n.s.
Former smokers, n (%)	11 (44)	11 (40.7)	n.s.
Asthma onset under 18 y, n (%)	9 (36)	13 (48.1)	n.s.
Asthma onset over 40 y, n (%)	9 (36)	7 (25.9)	n.s.
Atopy, n (%) PAS, n (%) SAS, n (%)	19 (76) 10 (40) 9 (36)	20 (74) 19 (70) 1 (3.7)	n.s. 0.027
Allergic rhinitis, n (%)	19 (76)	17 (62.9)	n.s.
Atopic dermatitis, n (%)	5 (20)	8 (29.6)	n.s.
CRSwNP, n (%)	20 (80)	10 (37)	0.001
GERD, n (%)	8 (32)	17 (62.9)	0.024
Obesity, n (%)	0 (0)	11 (40.7)	0.0003
ACOS, n (%)	1 (4)	2 (7.4)	n.s.
OSAS, n (%)	1 (4)	6 (22.2)	0.043
Emphysema, n (%)	0 (0)	3 (11.1)	n.s.
Bronchiectasies, n (%)	1 (4)	6 (22.2)	0.043
Hypertension, n (%)	1 (4)	8 (29.6)	0.014
Type 2 diabetes mellitus, n (%)	2 (8)	2 (7.4)	n.s.
ACT (pts)	15 ± 6	18 ± 5	n.s.
AE rate (ep/months)	0.16 ± 0.20	0.15 ± 0.23	n.s.
Severe AE (n)	0.16 ± 0.473	0.52 ± 0.75	n.s.
FVC (lt)	3.961 ± 1.323	3.311 ± 1.135	0.043
FEV1 (lt)	2.739 ± 0.804	2.011 ± 0.842	0.004
FEF_25–75_ (lt)	1.979 ± 1.043	1.036 ± 0.651	0.002
FEV1/FVC	70 ± 8.9	59 ± 12.2	0.0007
Blood eosinophils (cell/µL)	581 ± 390	332 ± 202	0.008
Total serum IgE (kU/L)	841 ± 1125	998 ± 1928	n.s.
FeNO > 25 ppb, n (%)	8 (88) *	6 (46) *	0.032
High-dose ICS, n (%)	20 (80)	24 (88)	n.s.
Manteniance OCS, n (%)	15 (60)	13 (48.1)	n.s.
Daily prednisone equivalence (mg)	3.4 ± 5.8	6.8 ± 9	n.s.
LAMA, n (%)	14 (56)	11 (40.7)	n.s.

Data are reported as mean ± standard deviation for continuous variables or absolute (relative) frequency for nominal variables. The comparison between baseline variables according to remission status at 12 months has been performed with *t*-test for continuous variables or chi-square test/Fisher’s exact test for nominal variables. n.s. = not significant. 2 SAS = exclusively seasonal allergen sensitization; PAS = perennial allergen sensitization; other abbreviations are reported in the Materials and Methods Section. * data available for 22 out of 52 patients.

**Table 3 biomedicines-13-02404-t003:** Multivariate logistic regression analysis.

Dependent Variable: Remission at 12 Months				
Independent Variables at Baseline	*p*-Value	OR	CI 95% Lower	CI 95% Upper
FEV1 (L)	0.026	2.941	1.13	7.6
CRSwNP (yes)	0.018	8.03	1.41	45.5
Blood eosinophils (cell/µL)	0.015	1.003	1.001	1.006
Bronchiectasies (yes)	0.322	0.217	0.011	4.47
Obesity (yes)	0.554	0.520	0.76	1.14
Perennial allergen sensitization (yes)	0.189	0.824	0.02	4.35

## Data Availability

The original contributions presented in the study are included in the article/Appendix A, further inquiries can be directed to the corresponding author.

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
