# Peer review of "Determinants of Clinical Remission in Dupilumab-Treated Severe Eosinophilic Asthma: A Real-World Retrospective Study"

_biomedicines, 2025, doi:10.3390/biomedicines13102404_

Round 1
Reviewer 1 Report
Comments and Suggestions for Authors
-The topic is interesting and the authors have done their best to broaden this important issues in the area of respiratory diseases, particularly asthma. But it needs revision for clarity and correction for this nice topic and work to be finally accepted in Biomedicines.
-Throughout the text, some very unproven concluding points like therapeutic ones throughout the text should be avoided; the study lack such results and it is advised to calming down such points in the revision.
-In the abstract please clearly mention the level of Eosinophils increased/decreased by dupilumab sig… (>490 cells/μl compared to healthy cohort)?
-In the text of the methods it is unclear which predictor(s) was/were key along with the unclear mentioning of the methods? It is also necessary to clearly mention some normal values of healthy counterparts? With also mentioning reference(s).
-I would add a clear depiction for the experiments design in the methods.
-Some units should be mentioned in a scientifically standard and correct manner. E.g. Please replace appropriate unit for ppb? cell/mcl? And more…; please elaborate.
-Throughout the text please use better abbreviation(s); e.g, for FeNO; readers might make mistake “Fe” for “Iron”?
In my opinion the conclusions (both in the abstract and at the end of the discussion) are too generic; they should be more clear and straightforward, and should report more specific information and summarize what was described in the predictors of the e.g., “key predictors of clinical remission”… those key ones should appear here.
-your key word should be more relevant; I would use “dupilumab”, “asthma” etc… and remove irrelevant ones..
-L 42, …Type 2 (T2)-high….?
-L 56, …. across other biologics:….? Please elaborate?
-In the text of the methods please clearly mention number of patients in the retrospective study design? The study population/demography? E.g., LL 101-111, should be removed from results part and moved to the methods section. I would also make a very informative table from the text LL, 101-119 and include in the methods section.
-For the caption of tables/figures please mention what is the main message of each? Also what is the n? The “n=?” should be addressed in the captions of the each figures and tables of the results.
The authors should comparatively address some points related to the Eosinophils and Key ILs mentioned here.
-L 146, … has been observed in pulmonary function between….?
-L149, …. A significant decalage of maintenance…?
-LL 165-166, … cohort, nor atopy, smoking history, neither a previous biological… unclear.
-LL 193-194, Grammatically unclear.
-Fig 3F, … where is R2? and P?
L 276, …..on the peripheral airways resulting in DE insufflation
L 347, ….As previously expressed,…?
LL 351-354, unclear.
-I would add/depict/shoe a figure with clear mechanistic scientific concepts, by including the Eosinophils, IL-4, IL-13 along with the dupilumab in the discussion part.
Apart from substantial unclarity, the study lacks addressing on clinically mechanistically aspects of very complex and multifactorial pathogenesis of asthma; at least addressing some key topical words aimed in this retrospective study.
Good luck
Comments on the Quality of English LanguageShould be improved.
Author Response
REVIEWER 1
The topic is interesting and the authors have done their best to broaden this important issues in the area of respiratory diseases, particularly asthma. But it needs revision for clarity and correction for this nice topic and work to be finally accepted in Biomedicines.
We sincerely thank the reviewer for the positive evaluation of our work and for the constructive comments. We appreciate the acknowledgment of the relevance of this topic in the field of respiratory diseases, and we have carefully revised the manuscript introducing the majority of his/her suggestions.
Throughout the text, some very unproven concluding points like therapeutic ones throughout the text should be avoided; the study lack such results and it is advised to calming down such points in the revision.
We thank the reviewer for this remark. We agree that some statements in the previous version were too strong, especially regarding therapeutic implications. We have carefully revised the manuscript and rephrased several sentences throughout the text, in particoular in the conclusions (L32-37; 263, 265, 271, 282, 389-391, 395), to present the findings in a more cautious and balanced way.
In the abstract please clearly mention the level of Eosinophils increased/decreased by dupilumab sig… (>490 cells/μl compared to healthy cohort)?
We thank the reviewer for the comment. We agree that eosinophil counts (as well as FeNO and IgE levels) should always be interpreted in the context of reference values from the general population. In our study, the threshold of 490 cells/µL represents a high value, as the upper limit of normal can be considered around 250 cells/µL (Hartl S, Eur Respir J 2020). Accordingly, we have modified the abstract by specifying “high” eosinophil levels (L25) and we have added the reference ranges in the manuscript (L134-135) and in the the Online Data Supplement.
In the text of the methods it is unclear which predictor(s) was/were key along with the unclear mentioning of the methods? It is also necessary to clearly mention some normal values of healthy counterparts? With also mentioning reference(s).
We thank the reviewer for the suggestions. We acknowledge that the Methods section of the main manuscript was not sufficiently detailed in this regard, as the complete description of the variables considered was provided in the Online Data Supplement for space reasons (L93). Following the reviewer’s recommendation, we have now added reference values for continuous variables in the general population to improve clarity (L134-135).
I would add a clear depiction for the experiments design in the methods.
We thank the reviewer for the comment. The study design and timing of assessments were already described in the Methods section. However, we agree that a visual representation could improve clarity. We have therefore added a flow-chart illustrating the study design (Figure 1).
Some units should be mentioned in a scientifically standard and correct manner. E.g. Please replace appropriate unit for ppb? cell/mcl? And more…; please elaborate.
We thank the reviewer for this remark. We have carefully revised the manuscript to ensure consistent use of standard scientific units. In particular, blood eosinophil counts are now reported as cells/µL (instead of cell/mcl). FeNO values remain expressed in parts per billion (ppb), and IgE in kU/L, which are standard reporting units in clinical and research practice.
Throughout the text please use better abbreviation(s); e.g, for FeNO; readers might make mistake “Fe” for “Iron”?
We thank the reviewer for the comment. Almost all the abbreviations we used are universally recognized in the medical field (including FeNO) and they are fully described in the “abbreviation section” (page 15). AE, PAS and SAS are the only acronyms that are not universally established, and therefore we specified their meaning in nearly all their occurrences in the figure legends and tables.
In my opinion the conclusions (both in the abstract and at the end of the discussion) are too generic; they should be more clear and straightforward, and should report more specific information and summarize what was described in the predictors of the e.g., “key predictors of clinical remission”… those key ones should appear here.
We thank the reviewer for this constructive comment. We agree that the conclusions were too generic. We have therefore revised both the Abstract and the Discussion to provide more specific and straightforward statements, explicitly summarizing the key predictors of clinical remission identified in our cohort (L32-37, 391-393).
your key word should be more relevant; I would use “dupilumab”, “asthma” etc… and remove irrelevant ones.
We thank the reviewer for the suggestion. However, we deliberately excluded from the keywords those terms that are already included in the title of the manuscript. We leave any decision on this point to the Editor.
-L 42, …Type 2 (T2)-high….?
We modified this point, thanks for the suggestion.
-L 56, …. across other biologics:….? Please elaborate?
We thank the reviewer for the suggestion. We rephrased the sentence replacing “across other biologics” with “are similar for all other biologics for severe asthma”
-In the text of the methods please clearly mention number of patients in the retrospective study design? The study population/demography? E.g., LL 101-111, should be removed from results part and moved to the methods section. I would also make a very informative table from the text LL, 101-119 and include in the methods
We thank the reviewer for the suggestion. According to the conventional structure of observational clinical studies, the number of patients included and the detailed demographic and clinical characteristics are usually presented in the Results section rather than in the Methods. Nevertheless, we still included the sample size (n=52) in the paragraph 2.1 (L68). We agree that a table with the descriptive statistics of the cohort could be informative, but due to space constraints (the manuscript already includes three tables and five figures) we preferred to report these data in the text. We remain available to discuss this possibility with the Editor.
-For the caption of tables/figures please mention what is the main message of each? Also what is the n? The “n=?” should be addressed in the captions of the each figures and tables of the results.
We thank the reviewer for the comment. In our study, all data (with the exception of FeNO, as already specified) were available for the entire cohort of patients (n = 52) at each follow-up visit. Therefore, we would consider it redundant to repeat “n = 52” in every caption. For more clarity we now specified it in the methods section (L96-98).
Conversely, we have chosen to add a specific title to each table and figure, according to reviewer suggestion.
The authors should comparatively address some points related to the Eosinophils and Key ILs mentioned here
We thank the reviewer for this suggestion. We have expanded the Discussion by comparatively addressing the roles of eosinophils and the IL-4/IL-13 axis in the pathogenesis of severe asthma. In particular, we highlight how eosinophils represent a measurable biomarker of type 2 inflammation, while IL-4 and IL-13 are upstream drivers of airway remodeling and inflammation, both directly and through the modulation of adhesion molecules (L299-301, 305-319, 325-331).
L 146, … has been observed in pulmonary function between….?
L149, …. A significant decalage of maintenance…?
LL 165-166, … cohort, nor atopy, smoking history, neither a previous biological… unclear.
LL 193-194, Grammatically unclear.
L 276, …..on the peripheral airways resulting in DE insufflation
L 347, ….As previously expressed,…?
LL 351-354, unclear.
We thank the reviewer for highlighting these linguistic issues/refuses. We have carefully revised the manuscript to correct the unclear or ungrammatical sentences mentioned (e.g., lines 146, 149, 165–166, 193–194) and improved the overall clarity of the text.
-Fig 3F, … where is R2? and P?
We thank the reviewer for this observation. R and p have been calculated and added to the figures, and the statistical method used for their calculation has been described in the Methods section of the Online Data Supplement.
-I would add/depict/shoe a figure with clear mechanistic scientific concepts, by including the Eosinophils, IL-4, IL-13 along with the dupilumab in the discussion part.
As suggested, we have added a new mechanistic figure (now Figure 5) illustrating the interplay between Th2/ILC2 cells, IL-4, IL-5, IL-13, eosinophils, and the inhibitory action of dupilumab on the IL-4Rα receptor. This schematic provides a visual summary of the immunological pathways discussed and clarifies the upstream vs downstream effects of cytokine blockade.
Apart from substantial unclarity, the study lacks addressing on clinically mechanistically aspects of very complex and multifactorial pathogenesis of asthma; at least addressing some key topical words aimed in this retrospective study.
We agree with the reviewer. We have revised the Discussion to better address the clinical and mechanistic aspects of asthma pathogenesis. In particular, we now emphasize the complex and multifactorial nature of severe asthma, highlighting the coexistence of type 2–high pathways (IL-4/IL-13 axis, eosinophilic inflammation) and non–type 2 mechanisms (such as obesity-related and neutrophilic inflammation).
Reviewer 2 Report
Comments and Suggestions for Authors
The manuscript by Matteo Bonato et al., entitled “Determinants of clinical remission in dupilumab-treated severe eosinophilic asthma: a real-world retrospective study”, aimed to evaluate clinical outcomes, remission rates, and predictive factors of remission in a real-life cohort of patients with severe eosinophilic asthma treated with dupilumab. The manuscript offers a valuable contribution to understanding the effectiveness of dupilumab in a real context. I recommend publication with minor revisions:
- The authors in their study include both naive patients and patients switched to dupilumab from another therapy; this leads to heterogeneity in the cohort and therefore a limitation, as patients who changed previous drug therapy could represent a subgroup with different clinical characteristics compared to naive patients. The authors are suggested to provide a comparative analysis of the basal characteristics of the two subgroups.
- In the 156-line results, the authors state that no significant differences were observed in blood eosinophil counts from baseline to follow-ups. This is in contradiction with the literature, which reports an increase in exosinophils during treatment with dupilumab, due to its mechanism of action. This can also be seen from the therapeutic indication in the summary of product characteristics, which reports “Dupixent is indicated in adults and adolescents 12 years and older as add-on maintenance treatment for severe asthma with type 2 inflammation characterized by raised blood eosinophils...”. The authors should therefore justify this atypical data they report.
- The non-responder patient group is represented by only 5 patients. The conclusions drawn on this subgroup, such as the association with obesity and BMI, cannot be considered significant and generalizable, precisely due to the small number of the sample. Authors are advised to present these results with more caution.
Author Response
The manuscript by Matteo Bonato et al., entitled “Determinants of clinical remission in dupilumab-treated severe eosinophilic asthma: a real-world retrospective study”, aimed to evaluate clinical outcomes, remission rates, and predictive factors of remission in a real-life cohort of patients with severe eosinophilic asthma treated with dupilumab. The manuscript offers a valuable contribution to understanding the effectiveness of dupilumab in a real context. I recommend publication with minor revisions:
We warmly thank the reviewer for the positive appraisal of our work and for the time and effort dedicated to the careful review of our manuscript. We truly appreciated the insightful comments and have accepted and implemented all the suggestions, which have helped us to improve the clarity and quality of the paper.
The authors in their study include both naive patients and patients switched to dupilumab from another therapy; this leads to heterogeneity in the cohort and therefore a limitation, as patients who changed previous drug therapy could represent a subgroup with different clinical characteristics compared to naive patients. The authors are suggested to provide a comparative analysis of the basal characteristics of the two subgroups.
We thank the reviewer for this important observation and we agree with the comment. We have therefore added a new table (Table E1, Online Data Supplement) comparing baseline characteristics of naïve patients and those switched from another biologic therapy. As shown, no significant differences were found between the two subgroups. This information has also been added to the Results section (L120-122).
In the 156-line results, the authors state that no significant differences were observed in blood eosinophil counts from baseline to follow-ups. This is in contradiction with the literature, which reports an increase in exosinophils during treatment with dupilumab, due to its mechanism of action. This can also be seen from the therapeutic indication in the summary of product characteristics, which reports “Dupixent is indicated in adults and adolescents 12 years and older as add-on maintenance treatment for severe asthma with type 2 inflammation characterized by raised blood eosinophils...”. The authors should therefore justify this atypical data they report.
We thank the reviewer for this comment. In our cohort, blood eosinophil counts almost doubled from V0 to V1, as previously reported in the literature. However, in our sample this difference did not reach statistical significance. This finding may be explained by the characteristics of our real-life population and by the retrospective design of the study, in particular the lack of standardized timing of blood sampling (e.g., consistent intervals, avoidance of proximity to OCS bursts). In addition, a relevant proportion of our patients (29%) were still receiving oral corticosteroids at V1, which may have attenuated the increase in blood eosinophil counts from V0 to V1. We have clarified this point in the manuscript (L174–177).
The non-responder patient group is represented by only 5 patients. The conclusions drawn on this subgroup, such as the association with obesity and BMI, cannot be considered significant and generalizable, precisely due to the small number of the sample. Authors are advised to present these results with more caution.
We thank the reviewer for the observation. As suggested, we have removed the section referring to obesity and BMI in the non-responder subgroup, given the very small sample size (n = 5), and acknowledge that no conclusions can be drawn. Accordingly, there are no remaining references to these factors in the Discussion.